# Learning from small data: Classifying sex from retinal images via deep learning

**Aaron Berk**[1]*, **Gulcenur Ozturan**[2], **Parsa Delavari**[2], **David Maberley**[3], **Özgür Yılmaz**[4], **Ipek Oruc**[2]

**1** Department of Mathematics & Statistics, McGill University, Montréal, Canada, **2** Department of Ophthalmology and Visual Sciences, University of British Columbia, Vancouver, Canada, **3** Department of Ophthalmology, University of Ottawa, Ottawa, Canada, **4** Department of Mathematics, University of British Columbia, Vancouver, Canada

* aaron.berk@mcgill.ca

**Data Availability Statement:** The ODIR dataset is available for download at https://odir2019.grand-challenge.org/dataset/. Retinal images sourced from Vancouver Coastal Health cannot be shared publicly due to patient confidentiality constraints.

## Abstract

Deep learning (DL) techniques have seen tremendous interest in medical imaging, particularly in the use of convolutional neural networks (CNNs) for the development of automated diagnostic tools. The facility of its non-invasive acquisition makes retinal fundus imaging particularly amenable to such automated approaches. Recent work in the analysis of fundus images using CNNs relies on access to massive datasets for training and validation, composed of hundreds of thousands of images. However, data residency and data privacy restrictions stymie the applicability of this approach in medical settings where patient confidentiality is a mandate. Here, we showcase results for the performance of DL on small datasets to classify patient sex from fundus images—a trait thought not to be present or quantifiable in fundus images until recently. Specifically, we fine-tune a Resnet-152 model whose last layer has been modified to a fully-connected layer for binary classification. We carried out several experiments to assess performance in the small dataset context using one private (DOVS) and one public (ODIR) data source. Our models, developed using approximately 2500 fundus images, achieved test AUC scores of up to 0.72 (95% CI: [0.67, 0.77]). This corresponds to a mere 25% decrease in performance despite a nearly 1000-fold decrease in the dataset size compared to prior results in the literature. Our results show that binary classification, even with a hard task such as sex categorization from retinal fundus images, is possible with very small datasets. Our domain adaptation results show that models trained with one distribution of images may generalize well to an independent external source, as in the case of models trained on DOVS and tested on ODIR. Our results also show that eliminating poor quality images may hamper training of the CNN due to reducing the already small dataset size even further. Nevertheless, using high quality images may be an important factor as evidenced by superior generalizability of results in the domain adaptation experiments. Finally, our work shows that ensembling is an important tool in maximizing performance of deep CNNs in the context of small development datasets.

Access to these data can be requested by contacting Sasha Pavlovich, Director of Data Access and Governance, Vancouver Coastal Health, at sasha.pavlovich@vch.ca. All other relevant data are already included within the body of the paper.

**Funding:** (AB) IVADO PostDoc-2022-4083608672 (AB) CRM Applied Math Lab postdoctoral funding (no number) (OY) NSERC Discovery Grant (22R82411) (OY) Pacific Institute for the Mathematical Sciences (PIMS) CRG 33 (IO) NSERC Discovery Grant (RGPIN-2019-05554) (IO) NSERC Accelerator Supplement (RGPAS-2019-00026) (IO & OY) UBC DSI Grant (no number) (IO) UBC Faculty of Science STAIR grant (IO & OY) UBC DMCBH Kickstart grant (IO & OY) UBC Health VPR HiFi grant Institut de valorisation des données https://ivado.ca Centres de recherches en mathématiques http://www.crm.umontreal.ca/labo/mathappli Natural Sciences and Engineering Research Council of Canada: https://www.nserc-crsng.gc.ca Pacific Institute for the Mathematical Sciences: https://www.pims.math.ca UBC DSI: https://dsi.ubc.ca/ UBC STAIR: https://science.ubc.ca/research/stair UBC DMCBH: https://www.centreforbrainhealth.ca/ UBC Health: https://health.ubc.ca/ The funders had no role in study design, data collection and analysis, decision to publish, or preparation of the manuscript.

**Competing interests:** The authors have declared that no competing interests exist.

## 1 Introduction

A retinal fundus reveals important markers of a patient's health [1, 2]. Retinal fundus photography (*retinal imaging*), in conjunction with manual image interpretation by physicians, is a widely accepted approach for screening patient health conditions such as: referable diabetic retinopathy, diabetic macular aedema, age-related macular degeneration, glaucoma, hypertension or atherosclerosis [1]. Recent investigations into the automated analysis of retinal images have suggested further improvements over manual interpretation, whether via increased efficacy, efficiency or consistency [1, 3–10]. In some cases, automated image analysis has been able to detect patient attributes that were previously thought not to be present or detectable in retinal images. For example, a deep learning (DL)-based algorithm for automated retinal image analysis, developed using an immense reference database of images, determined patient sex from retinal images with impressive efficacy (AUC = 0.97) [7].

DL approaches to automated retinal image analysis are seeing increasing popularity for their relative ease of implementation and high efficacy [4, 5, 7, 8]. DL is a kind of machine learning in which one or more neural networks, each having several *layers*, is *trained* on a dataset so as to approximately minimize its misfit between the model predictions and labels, typically using a stochastic training procedure such as stochastic gradient descent [11–13]. In the DL setting, for a classification task, a neural network is a highly overparameterized model (relative to the complexity of the *learning task* and/or dataset size) that has been constructed as a composition of alternately affine and non-linear transformations (*i.e.*, layers). This neural network *architecture* is designed with the intent of modelling dataset features at different scales, and several works have highlighted neural networks' efficacy for modelling collections of natural images [14–19].

In the work by Poplin et al. [7], large DL models were used to classify sex and other physiological and/or behavioural traits that were linked to patient health based on retinal fundus images. The classifiers were trained using Google computing services using a database of approximately 1.5 million images. In Gulshan et al. [4], an automated DL algorithm was compared against manual grading by ophthalmologists for detecting diabetic retinopathy (DR) in retinal images. The algorithm, based on the Inception-V3 architecture [20], was trained on approximately 128000 images, and scored high sensitivity and specificity on two independently collected validation sets. For example, they achieved a (sensitivity, specificity) of (90.3%, 98.1%) on one validation set when tuning for high specificity; (97.5%, 93.4%) on the same validation set when tuning for high sensitivity. Results of Ting et al. [9] show comparable efficacy, and demonstrate adaptivity of the algorithm to datasets from multi-ethnic populations. The algorithm achieved an AUC of 0.879 for detecting referable DR. The authors used an adapted VGGNet network [19] as the model architecture. Another study trained a DL CNN using over 130000 color fundus images to detect of age-related macular degeneration [21]. For further review of DL applications to ophthalmology and retinal imaging see [1, 2, 8, 22, 23].

Despite their ease of use, deep neural networks can be notoriously difficult to train [24–27]. The high efficacy of automated approaches that was touted above is typically achieved only with massive amounts of data [4, 7]. For example, in a work advertising its use of a "small database", Choi et al. [3] required 10000 images to achieve 30.5% accuracy on a 10-class classification problem. Generally, it is unknown when DL models are able to achieve high performance on small datasets. For example, theoretical sample complexity bounds guarantee generalization performance for certain neural network architectures [25, 28], but these bounds are known to be loose [29]. Likewise, the assumptions on the training algorithm used to guarantee the generalization error bounds are untenable due to non-convexity of the learning task. Thus, a relationship between minimal dataset size and DL model performance is uncharacterized.

Moreover, it is possible this relationship cannot be characterized in general due to the aforementioned non-convexity and *intrinsic complexity* of a DL task on a particular dataset. In addition to difficulties manifest in achieving high efficacy on a classification task on a dataset using deep neural networks, there are further difficulties in guaranteeing the *stability* of the resultant trained model [25, 30, 31]. For example, if an automated method for image analysis was designed to achieve high performance on a homogenous population of retinal images, how is its performance affected when evaluated on a heterogeneous population of images? Finally, interpretability is a major challenge for DL models. Though the model of Poplin et al. [7] can classify sex with high efficacy, it is challenging to determine the underlying elements of an image that determine the network's classification. For this reason, a subsequent work [32] used classical methods (manual feature engineering and logistic regression) to design an interpretable classifier (AUC = 61.5%). An alternative approach retains the high-efficacy DL model, and creates *ex post facto* classifier interpretations using standard DL tools like class activation maps [33] or saliency maps [34]. Each is an integral component of a subsequent work by a subset of the present authors [35], in which a framework is developed for explainable classification of sex from fundus images by DL models.

Availability of very large, accurately labeled, high quality retinal imaging datasets with associated meta-data is limited [36]. Access to such existing datasets is further complicated by legal, ethical, technical, and financial barriers. An open challenge, crucial for widespread applicability, democratization, and generalization of DL in medical imaging is regarding whether high performance can be achieved with DL methods on challenging image classification tasks when the image database is small. Some work exists in this setting of so-called "low-shot" learning, but with different imaging modality (namely OCT) [37] or for a relatively easier classification task (namely DR) [38].

In this study, we take a step toward examining this open question by developing and evaluating a DL model to classify patient sex from retinal images for small data set sizes. The DL model architecture is a state-of-the-art ResNet [15] architecture, whose weights have been pretrained on the ImageNet database [39]. The pre-trained network is fine-tuned for classifying sex on a database of approximately 2500 fundus images. Furthermore, we investigate the stability of a neural network trained on a small database of images by testing its classification performance on out-of-distribution data. We approach this goal empirically through domain adaptation experiments, in which the performance of DL models trained on one dataset of retinal fundus images are then evaluated on another. Finally, we contrast the results of the models in this work with an enesemble model created by averaging scores of several individual models trained on small data sets.

## 2 Methods

### 2.1 Annotated retinal fundus image datasets

In this work we introduce two novel private datasets of retinal fundus images that have been curated from the Vancouver General Hospital ophthalmic imaging department: labeled DOVS-i and DOVS-ii datasets for the remainder of this paper.

The DOVS-i dataset was created from a total of 2025 images from 976 patients. Specifically, it was created by randomly subsampling this collection of images to include only a single pair (one left and one right) of retinal images from each patient, resulting in a collection of 1706 images from 853 patients. The DOVS-ii dataset was curated from a collection of 3627 images from 1248 patients. As with DOVS-i, the DOVS-ii dataset was created by randomly subsampling the image collection to include only a single pair of images from each patient: 2496 images from 1248 patients. For statistics regarding the make-up of each dataset, refer to

Table 1a and 1b, respectively. See section 4.4 for for detailed statistics (including data set age distribution). Note that since DOVS is composed of images taken for clinical purposes, ethnicity information is unavailable.

In this work we additionally use a publicly available dataset for which the retinal images contained therein were annotated with patient sex information. The so-called ODIR database of images [40] comprises 7000 images from 3500 patients. This database of images was subsampled, by us, to create the ODIR-N and ODIR-C datasets. The ODIR-N dataset is a subset of ODIR, containing only "normal" eyes (eyes without any annotated abnormality). The ODIR-N dataset contains 3098 images from 1959 individuals. The ODIR-C dataset is a subset of the ODIR-N dataset, which was subsampled by an ophthalmologist (Dr. Ozturan) to eliminate poor quality images, as image quality may contribute to classifier performance (*e.g.*, [41]). Given there are no clinical features in a retinal fundus image relevant to sex classification, quality assessment focused on ensuring that, as in standard fundus photography, the macula is in the centre of image, and the optic disc is located towards the nose, and that any extreme imaging artifacts were absent. Images with poor focus and illumination were also removed. The ODIR-C dataset contains 2577 images from 1722 patients. For statistics regarding the make-up of each dataset, refer to Table 1c and 1d, respectively; see section 4.4 for detailed statistics (including data set age distribution), and for criteria followed for the image elimination process. Note that since ODIR is composed of images taken for clinical purposes, ethnicity information is unavailable. Throughout this work, we may refer to the size of a partition using the notation $n_{\text{partition}}$. For instance, the size of the validation partition of the DOVS-i dataset is $n_{\text{val}} = 214$.

## 2.2 DOVS dataset curation and pre-processing

The corpus of DOVS images is comprised of color retinal fundus photographs obtained from the Vancouver General Hospital Ophthalmic Imaging Department. Individuals were randomly sampled from this database—without regard to age, sex or health status. For each

**Table 1. Dataset statistics.** Counts are given for the total number of images and patients, and for each of the train, validation and test sets. In addition, counts for the number of female and male images/patients in each of the datasets, and in aggregate, are included.

| | DOVS-i | | | | DOVS-ii | | | |
|---|---|---|---|---|---|---|---|---|
| | train | val | test | total | train | val | test | total |
| images | 1280 | 214 | 212 | 1706 | 1746 | 374 | 376 | 2496 |
| ↪ female | 656 | 112 | 108 | 876 | 888 | 190 | 192 | 1270 |
| ↪ male | 624 | 102 | 104 | 830 | 858 | 184 | 184 | 1226 |
| patients | 640 | 107 | 106 | 853 | 873 | 187 | 188 | 1248 |
| ↪ female | 328 | 56 | 54 | 438 | 444 | 95 | 96 | 635 |
| ↪ male | 312 | 51 | 52 | 415 | 429 | 92 | 92 | 613 |
| (a) DOVS-i | | | | | (b) DOVS-ii | | | |
| | ODIR-N | | | | ODIR-C | | | |
| | train | val | test | total | train | val | test | total |
| images | 2170 | 470 | 458 | 3098 | 1816 | 381 | 380 | 2577 |
| ↪ female | 980 | 209 | 209 | 1398 | 829 | 170 | 173 | 1172 |
| ↪ male | 1190 | 261 | 249 | 1700 | 987 | 211 | 207 | 1405 |
| patients | 1371 | 294 | 294 | 1959 | 1205 | 258 | 259 | 1722 |
| ↪ female | 631 | 135 | 135 | 901 | 553 | 118 | 119 | 790 |
| ↪ male | 740 | 159 | 159 | 1058 | 652 | 140 | 140 | 932 |
| (c) ODIR-N | | | | | (d) ODIR-C | | | |

individual sampled from the database, a single pair of images (left and right eye) was added to the data set. The individuals were assigned a unique token; this token, the sex of the individual and a left/right identifier were used to label each image. We describe each dataset below; see section 4.4 for more detailed data statistics.

**DOVS-i dataset.** A total of 853 patients were selected from the database, yielding a data set of 1706 images. The resolution of each retinal fundus image was $2392 \times 2048$ pixels. Before training, each image was resized to $300 \times 256$ using the Haar wavelet transform. Specifically, let $\mathcal{W}$ denote the Haar wavelet transform and let $x$ denote a retinal fundus image. Let $w := \mathcal{W}x$ so that $w_j$ is a tensor of the $j$th level of detail coefficients, where $j = 0$ corresponds to the coarsest level scaling coefficients. We remove the finest 3 scales by projecting onto the first $W - 3$ levels, and inverse-transform $\tilde{w} := (w_j)_{j=0}^{W-3}$ via $\tilde{x} = \mathcal{W}^{-1}\tilde{w}$ to obtain the appropriate down-sized image.

Individuals were randomly partitioned into training, validation and test sets in a way that maximally resembled the proportion of females to males in the aggregate population. This also ensured that both left and right eyes of an individual were always retained within the same partition. The training set was comprised of 640 patients (*i.e.*, $\sim 75\%$ of the data at 1280 images), the validation set of 107 patients ($\sim 12.5\%$ of the data at 214 images), and the test set of 106 patients ($\sim 12.5\%$ of the data at 212 images). Counts for each partition, including counts stratified by sex, may be found in Table 1a.

**DOVS-ii database.** The procedure used for the DOVS-ii dataset was similar to that for the DOVS-i dataset. The main difference is in the counts—of images and patients. A total of 1248 patients were selected from the database, yielding a dataset of 2496 images. The set of unique patients included in DOVS-ii is a superset of DOVS-i, however, since image selection for each patient was based on random sampling of multiple existing images for both DOVS-i and DOVS-ii, this is not true for the two image sets. The original size of each image is consistent with those for the DOVS-i dataset; each was again subsampled using the aforementioned Haar wavelet transform. The partitioning of the dataset into training, validation and test sets was 70%/15%/15%, retaining both left and right fundus images of each individual within the same partition. Counts for each partition, including counts stratified by sex, may be found in Table 1b.

**Ethics statement.** This is a retrospective study of archived samples. The study was approved by the UBC Clinical Research Ethics Board (H16–03222) and Vancouver Coastal Health Research Institute (V16–03222), and requirement of consent was waived.

## 2.3 Network architecture

The architecture of the deep learning model used in this work was modified from a deep residual network, ResNet-152 [15]. To our knowledge, the application of this network architecture to automated retinal image analysis tasks has not previously been explored, and neither for the classification of patient sex. Leveraging the common deep learning tool of transfer learning [42–46], the weights of the network are initialized using the original weights [15] learned for the ILSVRC'15 ImageNet image classification contest [39]. The final layer of the network was replaced by a new layer suitable for a binary classification task such as classifying patient sex. In our implementation of the transfer learning approach, during training the model weights are "fine-tuned", such that all weights in the network are updated. Updates were performed using stochastic gradient descent. For more details on training implementation and model architecture, see S1 Methods and S1 Parameter values in S1 File.

## 2.4 Network training

A "fine-tuning" [47] approach was used to train the network. In this framework, all weights of the network are allowed to vary, with the intuition that important features from the pre-

trained layers will vary little during training, having already reached approximate convergence for a large class of natural images. The expectation, then, is that the as yet untrained weights in the final fully connected layers will learn good features using the layer outputs from the rest of the network.

The criterion determining such a goodness of fit was chosen to be binary cross-entropy (BCE) loss. Specifically, BCE loss measures how well the network performs on the task of labeling the training data. Let $x$ denote a particular data point and suppose $h(x)$ takes on values between 0 and 1 representing the "belief" that x belongs to class 0 or 1. For example, $h(x) = 0$ (or $h(x) = 1$) would mean that we believe $x$ belongs to class 0 (or class 1) with certainty whereas intermediate values correspond to uncertain beliefs. With this in mind, the BCE loss $\ell(h(x), y)$ represents the level of "surprise", or discrepancy, between the true label $y$ and $h(x)$, namely:

$$\ell(h(x), y) := -y \log h(x) - (1 - y) \log (1 - h(x)).$$

Observe that $y \in \{0, 1\}$ and so exactly one of the above terms is non-zero for any image-label pair $(x, y)$ in the database. Furthermore, by convention we set $0 \log 0 = 0$.

The network was trained for a maximum of 300 passes through the training data, so-called epochs. On each epoch, the training algorithm was fed consecutive batches of the training data, from which approximate gradients of the loss function were computed to update the weights of the network. A batch size of 8 was used.

For each epoch, the order of the images was randomized. An image is loaded only once per epoch. For each image in each batch in each epoch, image augmentation was performed such that the image was randomly rotated with probability 0.2 by a random angle between −10 and 10 degrees inclusive; randomly cropped to a size of 224 × 224 pixels; a horizontal flip was applied with probability 0.3; histogram equalization was applied; and then the image was normalized using the standard mean and standard deviation parameters expected by ResNet:

$$\mu_{(r,g,b)} = [0.485, 0.456, 0.406] \qquad \sigma_{(r,g,b)} = [0.229, 0.224, 0.225].$$

The network was validated after each epoch on the validation set to determine both the validation loss and accuracy. Early stopping was employed to mitigate overfitting of the model during training. Specifically, model training was run for at least 3 epochs and effectively halted after validation accuracy ceased to improve. Early stopping then selected the final model for that model run to be the one from the epoch with highest-attained validation accuracy. For background on early stopping, see [48] or [49 Chapter 11.5.2]. Properties of the validation sets within each database are available in Table 1.

During a model run, several metrics were recorded to track the progress of the training procedure. In particular, accuracy (proportion correct), BCE and area under the curve of the Receiver Operating Characteristic (AUC) were measured after each epoch for the model on both the training and validation partitions of a dataset. A more detailed description of these three metrics may be found in S1 Methods in S1 File.

For each of the databases DOVS-i, DOVS-ii, ODIR-N, and ODIR-C, several model runs were performed—that is, several models were trained using the above described training procedure. In particular, 5 models were trained using the DOVS-i database; 6 each for the ODIR-N and ODIR-C databases. The models trained using the DOVS-ii database were used for a model ensembling experiment. A description of the development procedure for those models is reserved to Ensembling method. Each trained model is labelled with an identifier: D1–5 for the models trained using the DOVS-i dataset, E1–20 for the models trained using the DOVS-ii dataset, N1–6 for the models trained using the ODIR-N dataset, and C1–6 for the models trained using the ODIR-C dataset. For sake of concision, we may refer to a model run

metonymously with the identifier associated to the trained model it produced (*e.g.*, "model run D1"). We disambiguate the two only where we feel it could otherwise lead to confusion.

## 2.5 Network evaluation

The performance of a trained network was evaluated by computing a quantity of interest on a "hold-out" set of data. In most cases, this hold-out set is simply the test partition from the same dataset whose train and validation partitions were used to train the model, and the quantity of interest is the AUC score of the model. These are the results presented in Single-domain results, and whose procedure is described here. In Domain adaptation results, where we present results of domain adaptation experiments, the dataset used for test-time evaluation is no longer the associated test partition, but an external dataset. A description of that procedure appears in Domain adaptation method.

For each test of a model from a run, the quantity of interest was that model's area under the receiving operator characteristic (AUC) on the test set. For more information on the receiver operating characteristic (ROC) and AUC, see S1 Methods in S1 File. Significance of the results was evaluated using bootstrapping, as described in S1 Bootstrap method in S1 File. A $(1 - \alpha)$-confidence interval was computed from $B = 1000$ bootstrap replicates where $\alpha = 0.05$; *p*-values for each test were computed and compared with an $\alpha = 0.05$ significance level after being adjusted for multiple comparisons using the Benjamini-Hochberg procedure (BH) [50]. Namely, one bulk adjustment for multiple comparisons was performed, including all significance tests performed in this work. As such, this adjustment includes tests for models D1–5, N1–6 and C1–6 presented in Single-domain results (*cf*. Table 2). Other tests included in the adjustment will be described in the appropriate sections. Throughout, a significance level of $\alpha = 0.05$ was used. To each trained model is associated two tests: its AUC score on the validation set, and that on the test set. In tests for significance, the AUC score was compared to a reference score corresponding to random chance: $\mu_{\mathrm{AUC}}^{\mathrm{ref}} := 0.5$.

It is worth noting that tests for significance of a model's AUC score on the training set were not performed. Models were only selected if they converged; effectively, if a model converged, its AUC score on the training set is expected to be large by construction. At any rate, the desideratum of the present work is a model's performance on out-of-sample data, which would be the key indicator of its performance in a clinical setting.

## 2.6 Domain adaptation method

In this experiment, we investigated the efficacy of our approach for *transductive* transfer learning [51], in which the distribution of retinal images underlying the test set is different from that for the training set. The ability of a trained model to perform domain adaptation in this manner provides an empirical understanding of the network's *stability* on the learning task: how its performance varies when the underlying data distribution is varied. The experimental set-up was similar to that described in Network training and Network evaluation. The key difference is that models developed using the training and validation partitions of one dataset were tested on another *entire* database. Namely, models D1–5 were developed using the DOVS-i training and validation partitions and tested on both the ODIR-N and ODIR-C databases; models E1–10 and E* were developed using the DOVS-ii training and validation partitions and tested on both the ODIR-N and ODIR-C databases; models N1–6 were trained using the training and validation partitions of the ODIR-N database and tested on the DOVS-ii database; models C1–6 were trained using the training and validation partitions of the ODIR-C database and tested on the DOVS-ii database. Results of these experiments may be found in Domain adaptation results, reported in Table 4. As above, the quantity of interest was the

**Table 2. Single-domain results for fine-tuned ResNet-152 models.** $B = 1000$, $\alpha = 0.05$. Unless explicitly stated, $p_{\text{empir}} \leq 10^{-3}$ for all significance tests prior to correction for multiple comparisons, and $p_{\text{adj}} \leq 1.1 \cdot 10^{-3}$ for all significance tests after adjustment for multiple comparisons using BH. Significant results are marked by an asterisk. Trend for significance ($0.05 < p < 0.1$) is marked by a number sign.

| run | epoch | AUC | | $CI_\alpha$ | | | | | |
|-----|-------|-----|-----|-----|-----|---|---|---|---|
| | | val | test | val | test | | | | |
| D1 | 109 | 0.81* | 0.69* | (0.75, 0.86) | (0.62, 0.76) | | | | |
| D2 | 37 | 0.81* | 0.61* | (0.76, 0.86) | (0.54, 0.68) | | | | |
| D3 | 46 | 0.81* | 0.69* | (0.74, 0.86) | (0.62, 0.76) | | | | |
| D4 | 60 | 0.81* | 0.68* | (0.75, 0.86) | (0.61, 0.75) | | | | |
| D5 | 73 | 0.81* | 0.67* | (0.75, 0.86) | (0.59, 0.74) | | | | |

(a) DOVS-i; $n_{\text{val}} = 214$, $n_{\text{test}} = 212$.

| run | epoch | AUC | | $CI_\alpha$ | | | | | |
|-----|-------|-----|-----|-----|-----|---|---|---|---|
| | | val | test | val | test | | | | |
| N1 | 43 | 0.65* | 0.62* | (0.60, 0.70) | (0.57, 0.67) | | | | |
| N2 | 47 | 0.66* | 0.61* | (0.61, 0.70) | (0.55, 0.66) | | | | |
| N3 | 36 | 0.64* | 0.62* | (0.60, 0.69) | (0.57, 0.66) | | | | |
| N4 | 36 | 0.66* | 0.61* | (0.61, 0.70) | (0.57, 0.67) | | | | |
| N5 | 21 | 0.67* | 0.62* | (0.61, 0.72) | (0.57, 0.67) | | | | |
| N6 | 66 | 0.65* | 0.62* | (0.60, 0.70) | (0.57, 0.68) | | | | |

(b) ODIR-N; $n_{\text{val}} = 470$, $n_{\text{test}} = 458$.

| run | epoch | AUC | | $CI_\alpha$ | | $p_{\text{empir}}$ | | $p_{\text{adj}}$ | |
|-----|-------|-----|-----|-----|-----|-----|-----|-----|-----|
| | | val | test | val | test | val | test | val | test |
| C1 | 16 | 0.63* | 0.60* | (0.58, 0.69) | (0.54, 0.65) | <0.001 | <0.001 | <0.0011 | <0.0011 |
| C2 | 4 | 0.63* | 0.60* | (0.58, 0.69) | (0.54, 0.66) | <0.001 | <0.001 | <0.0011 | <0.0011 |
| C3 | 5 | 0.63* | 0.58* | (0.56, 0.68) | (0.52, 0.63) | <0.001 | 0.007 | <0.0011 | 0.0071 |
| C4 | 30 | 0.63* | 0.60* | (0.57, 0.69) | (0.55, 0.66) | <0.001 | 0.001 | <0.0011 | 0.0011 |
| C5 | 5 | 0.62* | 0.55# | (0.56, 0.67) | (0.49, 0.61) | <0.001 | 0.055 | <0.0011 | 0.0550 |
| C6 | 5 | 0.65* | 0.58* | (0.59, 0.70) | (0.52, 0.64) | <0.001 | 0.006 | <0.0011 | 0.0062 |

(c) ODIR-C; $n_{\text{val}} = 381$, $n_{\text{test}} = 380$.

AUC score of the model on the testing database. Significance of the results was evaluated using bootstrapping, and each collection of 22 tests (see Table 4a–4d, respectively) was adjusted for multiple comparisons using BH, as described in Network evaluation. As above, $(1 - \alpha)$-confidence intervals were computed using bootstrapping techniques as described in S1 Bootstrap method in S1 File.

## 2.7 Ensembling method

In Network architecture we describe how the models examined in this work are deep neural networks with a particular architecture, which map images to binary labels corresponding to the sex of the corresponding patient. In this section, we describe an additional experiment, in which a collection of such models are combined to form a single meta-classifier. The aim of these experiments is to understand the impact on classification performance of combining several stochastically trained models. For a review of ensembling, see [25, 49].

Fix numbers $\ell, L \in \mathbb{Z}$ satisfying $1 \leq \ell \leq L$. Fix a collection of trained models $f_i$, $i = 1, \ldots, L$, which take a retinal image as input, and output class probabilities (*e.g.*, the probability that the sex of the patient corresponding to the image is "male"). Without loss of generality assume that the models are ordered according to their validation AUC scores such that $AUC(f_1) \geq AUC(f_2) \geq \ldots \geq AUC(f_L)$. We define the $(\ell, L)$-ensemble of a collection of models as the

classifier whose output probabilities are the average of the output probabilities of the best $\ell$ models:

$$f^*(x) := \ell^{-1}\sum_{i=1}^{\ell} f_i(x).$$

In this work, we investigate the performance of a (10, 20)-ensemble classifier constructed from ResNet-152 models that were developed using the DOVS-ii dataset. These models correspond with $E1 - 20$, which were introduced in Network training. Efficacy of the ensemble was evaluated using the AUC score of the ensemble on the test partition of the DOVS-ii database. Significance of the scores (for the individual models and the ensemble) was determined using bootstrapping techniques described in S1 Bootstrap method in S1 File, including the determination of confidence intervals for the test AUC score at an $\alpha = 0.05$ confidence level. The AUC scores were compared to the reference null mean for AUC of $\mu_{\text{AUC}}^{\text{ref}} = 0.5$. As described previously, we adjusted for the multiple comparisons presented in this work using the BH procedure. Specifically, the $11 \times 2$ significance tests related to the model ensembling experiment (*cf.* Table 3) were included in the bulk adjustment described in Network evaluation.

## 2.8 Visual explanations for classification performance

To visually understand the performance of the learned classifiers, we use Guided Grad-CAM [33] to compute visual explanations of the regions in an image that most determine a model's score for that image. Broadly speaking, Guided Grad-CAM combines guided backpropagation [52] with a generalization of class activation maps (CAMs) [53]. Roughly, these visual explanations convey the regions of an image that most activate those filters that determine a particular class membership. In the present case, we supplement the efficacy of our DOVS-ii classifiers with visual explanations that highlight how the classifiers "focus" on physiologically relevant regions of a fundus image, such as the fovea, the vasculature and the optic disk (see Visual explanations for DOVS-ii classifiers). For this part of the work, we trained six new DOVS-ii classifiers to discriminate sex using an identical protocol to the one described for the previous

**Table 3. Ensembling experiment results: Scores for E1–10 developed using the DOVS-ii database, and the ensemble classifier $E^*$.** The unadjusted $p$-value obtained from bootstrap replicates satisfies $p_{\text{empir}} \leq 10^{-3}$ for each line in the table. The adjusted $p$-value $p_{\text{adj}}$ satisfies $p_{\text{adj}} \leq 1.1 \times 10^{-3}$ ($B = 1000$, $\alpha = 0.05$, $n_{\text{test}} = 376$). The row for run $E^*$ displays statistics for the (10, 20)-ensemble classifier created from the component models E1–10. Significant results are marked by an asterisk.

| run | epoch | AUC | | CI$_\alpha$ | |
|---|---|---|---|---|---|
| | | val | test | val | test |
| E1 | 45 | 0.76* | 0.71* | (0.71, 0.81) | (0.66, 0.76) |
| E2 | 34 | 0.76* | 0.69* | (0.71, 0.81) | (0.63, 0.74) |
| E3 | 63 | 0.75* | 0.71* | (0.70, 0.80) | (0.66, 0.77) |
| E4 | 13 | 0.75* | 0.70* | (0.70, 0.79) | (0.65, 0.75) |
| E5 | 6 | 0.75* | 0.67* | (0.70, 0.79) | (0.61, 0.72) |
| E6 | 49 | 0.75* | 0.70* | (0.70, 0.80) | (0.65, 0.75) |
| E7 | 10 | 0.74* | 0.69* | (0.69, 0.79) | (0.63, 0.74) |
| E8 | 17 | 0.74* | 0.69* | (0.69, 0.79) | (0.64, 0.74) |
| E9 | 21 | 0.74* | 0.68* | (0.70, 0.79) | (0.63, 0.74) |
| E10 | 11 | 0.74* | 0.69* | (0.69, 0.79) | (0.63, 0.74) |
| E* | — | 0.79* | 0.72* | (0.74, 0.84) | (0.67, 0.77) |

**Table 4. Domain adaptation results.** $B = 1000$, $\alpha = 0.05$. Unless explicitly stated, the unadjusted $p$-values satisfy $p_{empir} \leq 10^{-3}$ for all significance tests prior to correction for multiple comparisons, and $p_{adj} \leq 1.1 \cdot 10^{-3}$ for all significance tests after adjustment for multiple comparisons using BH. Significant results are marked by an asterisk.

| run | epoch | AUC | | $CI_\alpha$ | |
|---|---|---|---|---|---|
| | | ODIR-N | ODIR-C | ODIR-N | ODIR-C |
| D1 | 107 | 0.555* | 0.563* | (0.535, 0.577) | (0.540, 0.585) |
| D2 | 37 | 0.565* | 0.572* | (0.544, 0.585) | (0.551, 0.594) |
| D3 | 46 | 0.569* | 0.572* | (0.550, 0.589) | (0.550, 0.594) |
| D4 | 60 | 0.558* | 0.555* | (0.537, 0.580) | (0.533, 0.577) |
| D5 | 73 | 0.562* | 0.566* | (0.541, 0.581) | (0.543, 0.588) |

(a) DOVS-i models; $n_{ODIR\text{-}N} = 3098$, $n_{ODIR\text{-}C} = 2577$.

| run | epoch | AUROC | | $CI_\alpha$ | |
|---|---|---|---|---|---|
| | | ODIR-N | ODIR-C | ODIR-N | ODIR-C |
| E1 | 45 | 0.567* | 0.580* | (0.549, 0.589) | (0.560, 0.604) |
| E2 | 34 | 0.578* | 0.590* | (0.560, 0.598) | (0.569, 0.612) |
| E3 | 63 | 0.559* | 0.569* | (0.540, 0.579) | (0.546, 0.591) |
| E4 | 13 | 0.535* | 0.542* | (0.514, 0.554) | (0.521, 0.564) |
| E5 | 6 | 0.571* | 0.569* | (0.550, 0.591) | (0.545, 0.590) |
| E6 | 49 | 0.553* | 0.575* | (0.532, 0.572) | (0.552, 0.596) |
| E7 | 10 | 0.558* | 0.557* | (0.538, 0.578) | (0.535, 0.580) |
| E8 | 17 | 0.549* | 0.569* | (0.529, 0.569) | (0.548, 0.589) |
| E9 | 21 | 0.540* | 0.553* | (0.521, 0.559) | (0.531, 0.575) |
| E10 | 11 | 0.537* | 0.556* | (0.518, 0.557) | (0.535, 0.579) |
| E* | | 0.562* | 0.580* | (0.542, 0.584) | (0.558, 0.603) |

(b) DOVS-ii models; $n_{ODIR\text{-}N} = 3098$, $n_{ODIR\text{-}C} = 2577$. For E4, $p_{empir} = 10^{-3}$.

| run | epoch | AUC | $CI_\alpha$ | |
|---|---|---|---|---|
| | | DOVS-ii | DOVS-ii | |
| N1 | 43 | 0.580* | (0.558, 0.604) | |
| N2 | 47 | 0.584* | (0.563, 0.606) | |
| N3 | 36 | 0.607* | (0.585, 0.628) | |
| N4 | 36 | 0.588* | (0.566, 0.610) | |
| N5 | 21 | 0.600* | (0.579, 0.621) | |
| N6 | 66 | 0.583* | (0.561, 0.606) | |

(c) ODIR-N models; $n_{DOVS\text{-}ii} = 2496$.

| run | epoch | AUC | $CI_\alpha$ | |
|---|---|---|---|---|
| | | DOVS-ii | DOVS-ii | |
| C1 | 16 | 0.606* | (0.586, 0.630) | |
| C2 | 4 | 0.574* | (0.553, 0.595) | |
| C3 | 5 | 0.616* | (0.593, 0.636) | |
| C4 | 30 | 0.617* | (0.595, 0.638) | |
| C5 | 5 | 0.579* | (0.557, 0.601) | |
| C6 | 5 | 0.606* | (0.584, 0.627) | |

(d) ODIR-C models; $n_{DOVS\text{-}ii} = 2496$.

set of DOVS-ii models. We randomly selected 10 DOVS-ii images from the test partition and generated Guided Grad-CAM images (which we hereafter refer to as GGCAMs for simplicity) using the implementation [54]. For ease of interpretability, we manually selected a total of four sets of GGCAMs (two female, two male; two left, two right) that had adequately high contrast when plotted using the method described below. This mix of random and manual selection

was used to mitigate ex post facto bias in selecting fundus images to highlight in this paper while simultaneously permitting us to show visually salient images that can be easily interpreted by the reader. Finally, in Visual explanations for DOVS-ii classifiers we present the mean GGCAM for each image-sex pair, where the mean was taken over the six GGCAMs generated for each model. In addition, we also present the mean GGCAM *amplitude* and its standard deviation (*i.e.*, the mean of the pixel-wise $\ell_2$ norm of the GGCAMs).

Because a GGCAM is a three-channel image (*i.e.*, red, green and blue channels) that corresponds with the magnitude of the model's gradient for a given model-image pair, its values lie in $(-\infty, \infty)$. Thus, we normalize them to the range [0, 1] so they can be interpreted as false-colour images. For each fundus image presented, there are $6 \times 2 = 12$ individual GGCAMs (see S1 GGCAM plots in S1 File). Each of these images $X_i$, $i = 1, 2, \ldots, 12$ was pixel-wise normalized via the map $x \mapsto (x + v)/2v$ where $v > 0$ corresponds to the largest absolute extrema of the 12 images:

$$v := \max_{i=1,\ldots,12} \max\{|\min_{j,k,\ell}(X_i)_{jk\ell}|, \max_{j,k,\ell}(X_i)_{jk\ell}\}.$$

Above ($j$, $k$, $\ell$) represents the row/column index and channel of a pixel, respectively (*e.g.*, $\ell = 1, 2, 3$ for R, G and B). Note that this normalization strategy results in 0 being mapped to $0.5 \in [0, 1]$ for all such images we present.

Each individual GGCAM amplitude is computed as the pixel-wise $\ell_2$ norm of the GGCAM, meaning these images are now one-channel images (grayscale). We refer to these images as GGCAM-amp. For each retinal image, sex and model tuple we compute a GGCAM-amp and average over the models to obtain the mean GGCAM-amp visualizations presented in Visual explanations for DOVS-ii classifiers. There, we also show the standard deviation (with respect to the six trained models).

To reduce the number of plots in the main body of the work, we defer the individual GGCAMs and GGCAM-amps to S1 GGCAM plots in S1 File.

## 3 Results

### 3.1 Training metrics

Training-time metrics on both the train and validation partitions track a model's performance during training, and determine the extent of success of the training procedure. For all three datasets used in training—DOVS-i, ODIR-N and ODIR-C—accuracy, cross-entropy loss and AUC score were recorded for each training epoch. The data corresponding to these three metrics were plotted graphically as a function of model epoch. In Fig 1, a collection of six plots stratifies the data for D1 through D5 (pertaining to DOVS-i) according to partition (training/validation) and each of the three metrics. Specifically, accuracy appears in row (a), BCE in (b) and AUC in (c). Each line within an individual plot corresponds to the data collected for a single model run. The five DOVS-i model runs (D1–5) were plotted together to ease the graphical interpretation of aggregate trends during training. Similar data for training time metrics pertaining to model runs using the ODIR-N data are depicted in Fig 2; ODIR-C data, in Fig 3. For each model run, the final epoch with plotted data varies due to the early stopping condition that halted model training. The dashed vertical lines in each figure correspond with that epoch's model weights that were selected as the final model for that model run. The vertical lines correspond with the plotted data according to the colour, as depicted in the legend.

In our experiments, we observed that convergence of the training accuracy corresponded with that of the training AUC; inversely so with the training cross-entropy loss. Further, we observed that the validation accuracy and AUC corresponded with their training set

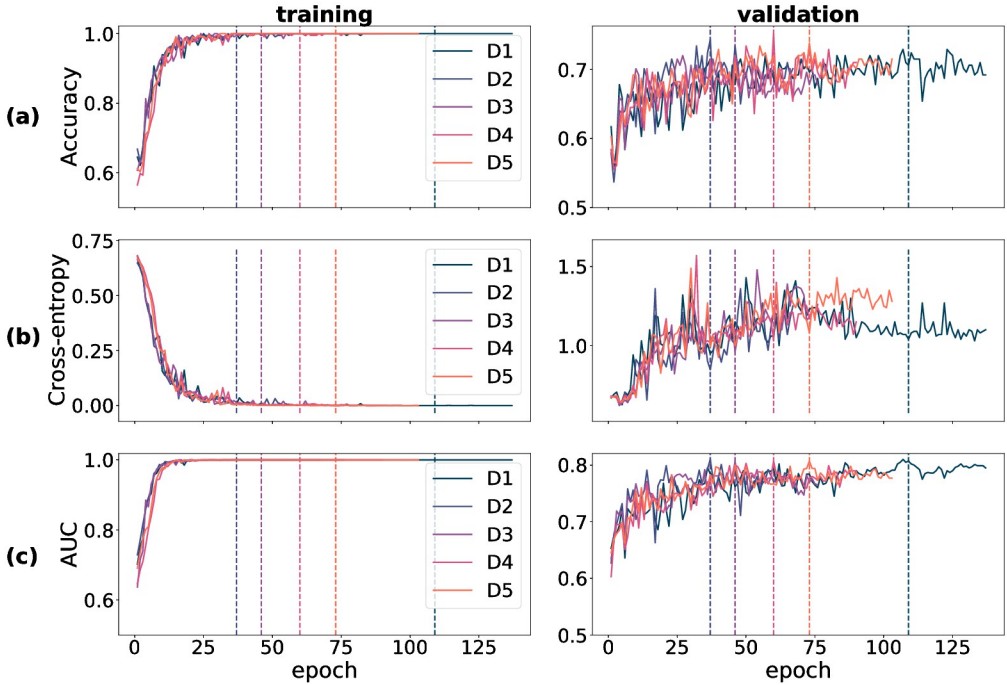

**Fig 1. Training-time metrics for runs D1 through D5.** (a) accuracy score (proportion correct); (b) binary cross-entropy loss; (c) AUC score. Vertical dashed lines correspond with the best epoch as selected by early stopping.

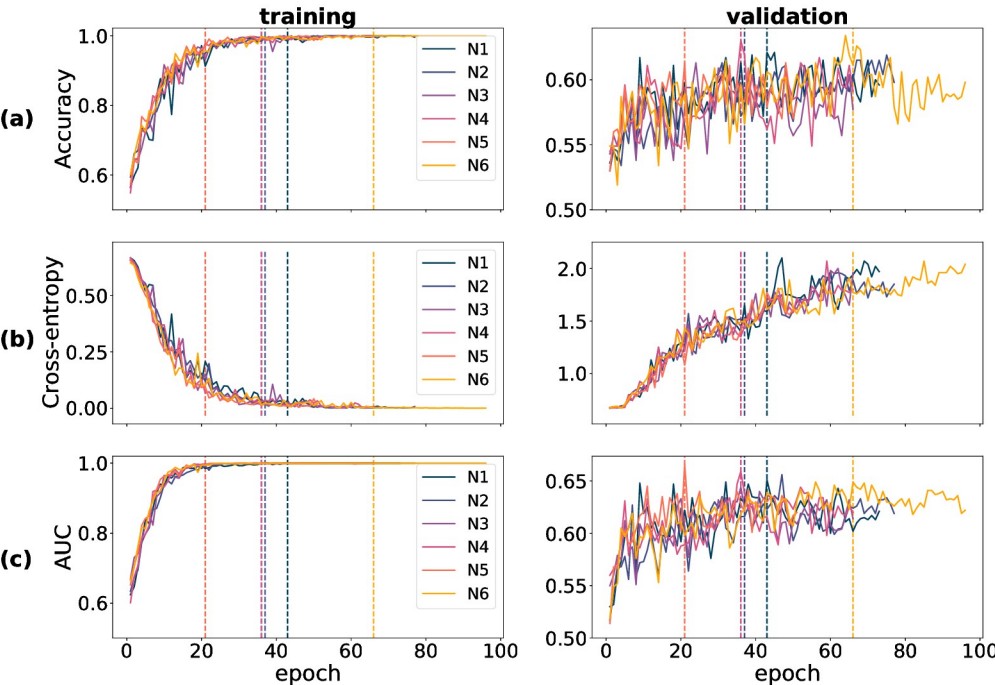

**Fig 2. Training-time metrics for runs N1 through N6.** (a) accuracy score (proportion correct); (b) binary cross-entropy loss; (c) AUC score. Vertical dashed lines correspond with the best epoch as selected by early stopping.

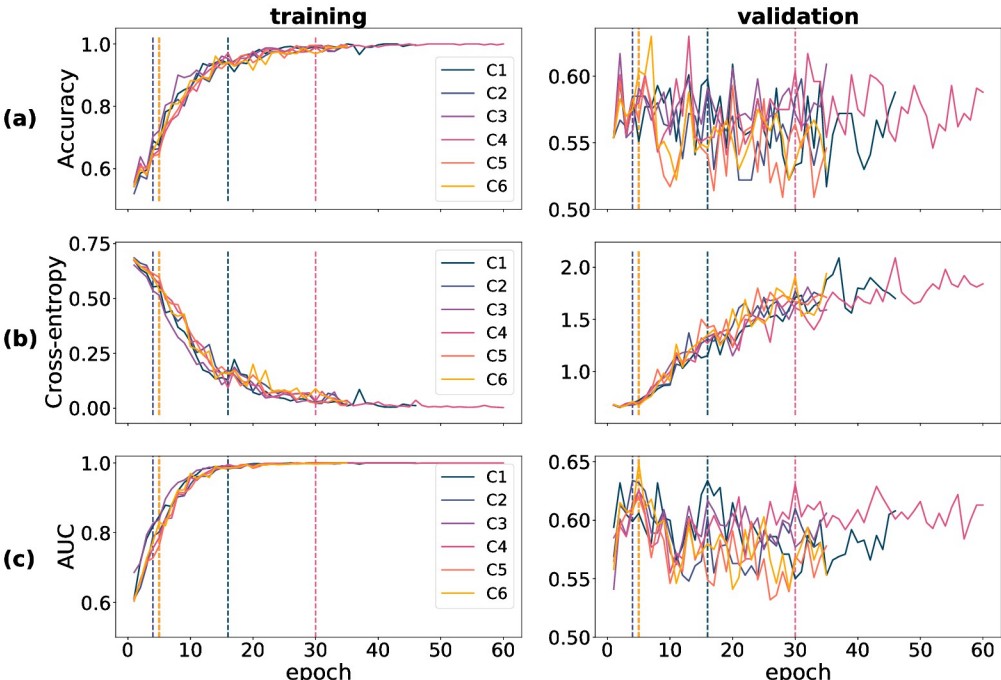

**Fig 3. Training-time metrics for runs C1 through C6.** (a) accuracy score (proportion correct); (b) binary cross-entropy loss; (c) AUC score. Vertical dashed lines correspond with the best epoch as selected by early stopping.

analogues, albeit with more noise and overall lower scores, as should be expected. (By "noise", here, we mean the variance of the training metric as a function of the run epoch.) In contrast, we observed that the cross-entropy loss behaved relatively erratically on the validation set. Notably, unlike that for the training partition, the validation cross-entropy loss *increased* for a majority of the epochs before stabilizing. This pattern is not inconsistent with the validation of a complex model trained on a small dataset. We conjectured that this behaviour comes about due to two phenomena: (1) the model's tendency to move towards more confident responses as training progresses, resulting in a few incorrect but confident responses coupled with (2) the imbalanced nature of BCE loss that imposes unbounded penalty to confident incorrect decisions. More specifically, the drop in the loss from one epoch to another when an uncertain incorrect decision is replaced with a a confident correct decision is much smaller compared to the increase in the loss when an uncertain incorrect decision is replaced with a confident incorrect decision. To test this hypothesis, we carried out an experiment in which we replaced BCE loss with a "balanced" loss function. Consistent with our expectation, we found that validation loss decreased in tandem with training loss when the loss function is balanced (see S1 Choice of loss in S1 File for methodological details and results of this experiment). It is worth mentioning here that although this supplementary experiment allows us to describe the conditions that bring about an increasing validation loss concurrently with improvements in other performance metrics, it does not explain why the model produces these few confident incorrect decisions. Such an account is beyond the scope of the present study.

For models trained using the DOVS-i data, convergence of the three metrics was typically attained within the first 25–75 epochs. Atypical in this instance is D1, which was obtained at epoch 109. We observe that convergence on the training data is attained faster than on the validation data; and that performance on the validation data may continue to improve even after

the scores on the training data are near-perfect. This observation is consistent with modern deep learning research in which training typically proceeds well after 0 training loss is attained.

For model runs training using the ODIR-N data, as with D1–5, the training metrics for N1–6 generally corresponded with another, except in the case of of validation BCE. We observed that validation accuracy was relatively noisier for N1–6 than for D1–5. For all six runs, model convergence was reached within 20–70 epochs, as depicted by the vertical dashed lines in each panel of Fig 2. Again we observe that convergence on the training data is attained more quickly than on the validation data; and that performance on the validation data may continue to improve even after the scores on the training data are near-perfect.

For model runs training using the ODIR-C data and yielding models C1–6, we observe that both validation accuracy and AUC appear to be markedly more erratic, compared to N1–6, as a function of model epoch. Given that ODIR-N already has few images relative to the complexity of the model architecture, it is unsurprising to observe more erratic and lower validation scores from training with ODIR-C, which is an even smaller dataset of images. Interestingly, all six models (C1–6) were selected by early stopping from within the first 30 epochs. This is indicative that the training procedure may be prone to overfitting on this especially small dataset. We reserve a more detailed comparison between DOVS-i and ODIR-C to Model-database duality.

## 3.2 Single-domain results

In the single-domain machine learning task, a trained and validated model is evaluated on an independent set of "test" data whose makeup resembles that of the training data. In this section we include scores, confidence intervals and significance values pertaining to experiments using a single database. We have computed these values for both the validation and test sets so that the validation performance may serve as a reference point. These values are reported in Table 2. Specifically, for models trained on the DOVS-i training set and evaluated on the DOVS-i validation and test sets, see Table 2a; for models trained on the ODIR-N training set and evaluated on the ODIR-N validation and test sets, see Table 2b; and for models trained on the ODIR-C training set and evaluated on the ODIR-C validation and test sets, see Table 2c. Finally, results for models trained on the DOVS-ii training set and evaluated on the DOVS-ii validation and test sets appear later in Table 3, as these models were used in the ensembling experiment whose results are presented in Ensembling.

The median test AUC score for the models D1–5 was 0.68. The validation AUC scores were all approximately 0.81, suggesting overfitting to the validation partition. The adjusted $p$-values $p_{adj}$ were computed using $B = 1000$ bootstrap replicates with sample sizes $n_{val} = 214$ and $n_{test} = 212$. For each of the tests for significance, $p_{adj}$ was found to satisfy $p_{adj} \leq 1.1 \cdot 10^{-3}$, implying the performance was significantly better than chance (0.5) at an $\alpha = 0.05$ confidence level. Table 2a shows 95% confidence intervals (CI) for the validation and test AUC scores for each of the five models (D1–5).

The median test AUC score for models N1–6 was 0.62, and the median validation AUC score was 0.66, suggesting that the extent of overfit was minimal. The adjusted $p$-values $p_{adj}$ were computed using $B = 1000$ bootstrap replicates with sample sizes $n_{val} = 470$ and $n_{test} = 458$. For each of the tests of significance, $p_{adj}$ was found to satisfy $p_{adj} \leq 1.1 \cdot 10^{-3}$, implying the performance was significantly better than chance (0.5) at an $\alpha = 0.05$ confidence level. Confidence intervals for the test AUC score for these models at the same confidence level were typically about (0.57, 0.67), all six nearly identical. All values for each of the six models (N1–6) are given in Table 2a.

The median test AUC score for the models C1–6 was 0.59, and the median validation AUC score was 0.63, suggesting the extent of overfit was minimal. The adjusted $p$-values $p_{\text{adj}}$ were computed using $B = 1000$ bootstrap replicates with sample sizes $n_{\text{val}} = 381$ and $n_{\text{test}} = 380$. For all tests for significance on the validation AUC, $p_{\text{adj}}$ was found to satisfy $p_{\text{adj}} \leq 1.1 \cdot 10^{-3}$, implying the validation scores were found to be significant at an $\alpha = 0.05$ confidence level. For all but one test for significance on the test AUC scores, $p_{\text{adj}}$ was found to satisfy $p_{\text{adj}} \leq 7.1 \cdot 10^{-3}$, implying all but one of the test scores were found to be significant at an $\alpha = 0.05$ confidence level. All values, including 95% confidence intervals, for each of the six models (C1–6) are given in Table 2a.

### 3.3 Ensembling

In this section, we describe results corresponding to the (10, 20)-ensemble classifier described in Ensembling method. In particular, we include in this section the results pertaining to the validation and test AUC performance of 10 models developed using the training and validation partitions of the DOVS-ii database, E1–10. The models E1–10 attained a median test AUC score of 0.69 and scores ranged between 0.67 and 0.71. The median validation AUC score was 0.75 and scores ranged between 0.74 and 0.76, suggesting negligible extent of overfit to the validation partition. All but one model run converged within the first 49 epochs, as displayed in the "epoch" column of Table 3. The ensemble classifier attained an AUC score of 0.72 on the test partition of the DOVS-ii database, higher than any test score for the constituent models. Its AUC score on the validation partition was 0.79, suggesting some level of overfit to the validation set during model development.

The adjusted $p$-values $p_{\text{adj}}$ were computed using $B = 1000$ bootstrap replicates with sample sizes $n_{\text{val}} = 214$ and $n_{\text{test}} = 212$. For each of the tests for significance, $p_{\text{adj}}$ was found to satisfy $p_{\text{adj}} \leq 1.1 \cdot 10^{-3}$, implying the results were found to be significant at an $\alpha = 0.05$ confidence level. Confidence intervals for the test AUC score for the 11 models were computed at the $\alpha = 0.05$ confidence level. Amongst the component models, the confidence intervals were similar. The confidence interval for the ensemble test AUC score was found to be (0.67, 0.77). All values are shown in Table 2a.

### 3.4 Domain adaptation results

In this section we describe results pertaining to the domain adaptation experiments, including scores, confidence intervals and significance values. All values pertaining to these experiments are given in Table 4. Specifically, for the AUC scores and associated significance scores of D1–5 evaluated on the ODIR-N and ODIR-C datasets, see Table 4a; for those pertaining to N1–6 evaluated on the DOVS-ii dataset, see Table 4c; and for those pertaining to C1–6 evaluated on the DOVS-ii dataset, see Table 4d.

When tested on the ODIR-N and ODIR-C datasets, models D1–5 attained a median AUC score of 0.564. AUC scores ranged between 0.555 and 0.572 and were all significantly greater than chance-level performance ($p_{\text{adj}} \leq 1.1 \cdot 10^{-3}$ for $B = 1000$ bootstrap replicates with $n_{\text{ODIR-N}} = 3098$ and $n_{\text{ODIR-C}} = 2577$). Confidence intervals for these results were computed at an $\alpha = 0.05$ confidence level and appear in Table 4a. The difference in scores between tests on ODIR-N and ODIR-C are minimal; for each model the confidence intervals associated to the two scores typically have a high degree of overlap.

When tested on the DOVS-ii dataset, models N1–6 attained a median AUC score of 0.586. AUC scores ranged between 0.580 and 0.607 and all were significantly greater than chance ($p_{\text{adj}} \leq 1.1 \cdot 10^{-3}$ for $B = 1000$ bootstrap replicates with $n_{\text{ODIR-N}} = 2496$). Confidence intervals for these results were computed at an $\alpha = 0.05$ confidence level and appear in Table 4a. When

tested on the DOVS-ii database, models C1–6 attained a median AUC score of 0.61. AUC scores ranged between 0.574 and 0.617 and all were significantly greater than chance ($p_{adj} \leq 1.1 \cdot 10^{-3}$ for $B = 1000$ bootstrap replicates with $n_{ODIR-N} = 2496$). Confidence intervals for these results were computed at an $\alpha = 0.05$ confidence level and appear in Table 4a. Despite being developed using a smaller dataset, C1–6 overall achieved greater test AUC compared to N1–6, though their performance is similar.

## 3.5 Model comparison

A summary overview of all models developed in the study is shown in Figs 4 and 5. In Fig 4 validation AUC is plotted against test AUC. For example, models D1–5 were developed using the training and validation partitions of the DOVS-i dataset; their AUC score on the test partition is plotted as a function of their AUC score on the validation partition. Only the retained subset (i.e., top 10) of E1–20 are depicted (labeled E1–10), as the other models were discarded. during development of the ensemble (*cf*. Ensembling method).

This plot enables a visual comparison of model performance on different datasets. In particular, it enables a direct comparison of models' test AUC scores, the variability of test AUC, and the extent of overfitting. For example, C1–6 generally attained the lowest validation and test scores, with N1–6 only slightly better. In particular, models performed better on the single domain task when developed using the DOVS datasets.

The ensemble E* attained the highest test score. Importantly, its validation score is less than that for D1–5, meaning that the extent of overfitting for E* is less. Its extent of overfitting is comparable with its constituent models E1–10. Indeed, except for models D1–5, we interpret that all models overfit by approximately 0.05. Finally, we observe that E1–10 have higher test AUC and overfit less than D1–5. In particular, the models developed with the larger DOVS-ii dataset performed better than those developed using DOVS-i.

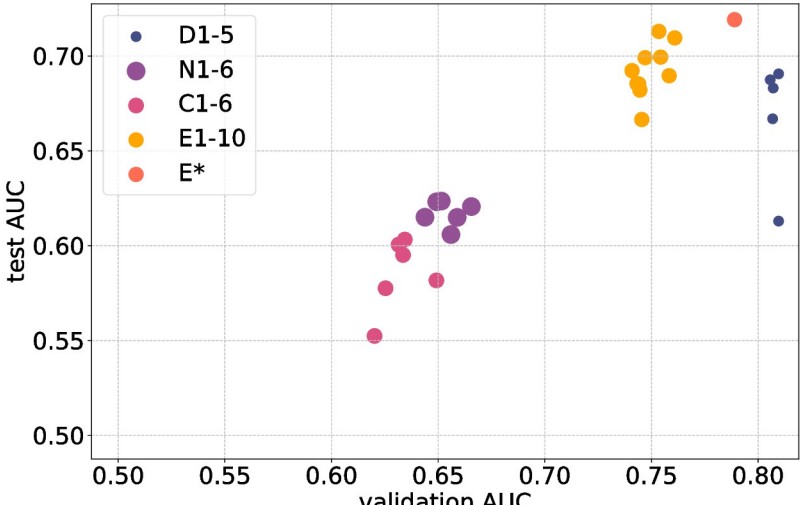

**Fig 4. A graphical representation of all models developed in this work.** On the *x*-axis is a model's AUC score on the validation partition of the relevant database; on the *y*-axis, its AUC score on the test partition. Points labelled D1–5 correspond with models trained and evaluated on DOVS-i; E1–10, DOVS-ii; N1–6, ODIR-N; C1–6, ODIR-C. The ensemble model is denoted E*. Marker size corresponds with the number of images in the database used for model development (*cf*. Table 1).

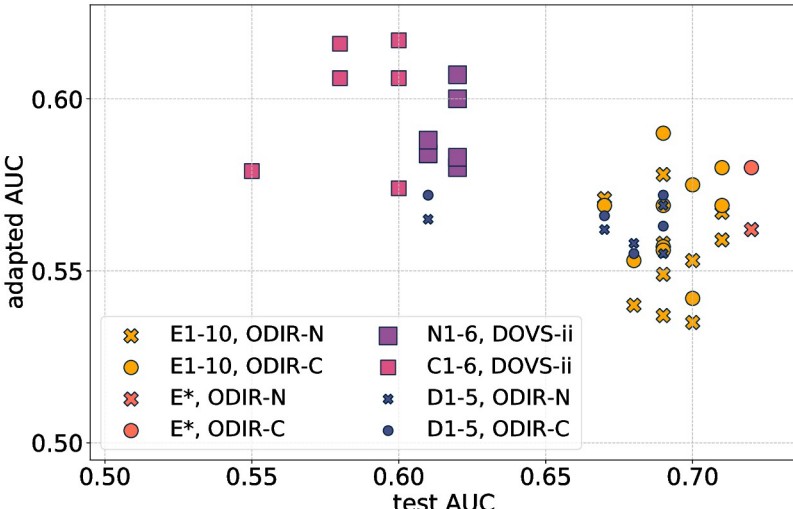

**Fig 5. A graphical representation of the domain adaptation results for the models developed in this work.** On the *x*-axis is a model's AUC score on the test partition from the database on which it was trained; on the *y*-axis, its domain-adapted AUC score. Points labelled D1–5 correspond with models trained and evaluated on DOVS-i; E1–10, DOVS-ii; N1–6, ODIR-N; C1–6, ODIR-C. The ensemble model is denoted E*. The marker radius of the plotted points corresponds with the number of images in the database used for model development (*cf.* Table 1). Each point is coloured according to the database on which the associated model was trained, and the shape of each point corresponds with the database on which that model was evaluated for its domain-adapted AUC score.

For those models for which there were multiple runs, we observe that the variability of the test AUC was comparable. Namely, the lowest and highest test AUC scores for a group of models was generally within 0.05, except one outlier each for D1–5 and C1–6.

In Fig 5, test AUC is plotted against adapted AUC, meaning the AUC score that was achieved on the domain adaptation task for the given (model, dataset) pair. Two clusters are discernible from the plot: the left cluster is comprised mainly of N1–6 and C1–6 models tested on DOVS-ii, who achieve lower test AUC scores (compared to the other discernible cluster), but adapt better to out-of-distribution data (*i.e.*, DOVS-ii). The other discernible cluster is comprised of models trained on DOVS-i or DOVS-ii. We observe that the ensemble model E* acheives the highest test AUC *score* and one of the highest adapted AUC scores on both ODIR-N and ODIR-C.

## 3.6 Visual explanations for DOVS-ii classifiers

We now use Guided Grad-CAM [33] to present visual explanations with the intent to understand the image regions that contribute to the classifers' score. As described in Visual explanations for classification performance, the images presented below are the mean GGCAMs and GGCAM-amps and the latter's standard deviation (for the individual GGCAM and GGCAM-amp images, and information pertaining to the normalization used for the colour images, see S1 GGCAM plots in S1 File). In each subfigure of Fig 6, these objects correspond with the left-most, centre and rightmost column, respectively. Thus, the images we present in this section showcase in aggregate the salient image regions for four individual fundus photos. The summary statistics were computed with respect to the six newly trained DOVS-ii models. The top row of the figure corresponds with two "Female" eyes, the bottom row, "Male". Colour bars are provided for each set of four grayscale images. Note that the pixel-wise standard deviation for the GGCAM images is not shown, because the resulting data range was insufficient to produce a visually meaningful plot (the images are essentially black).

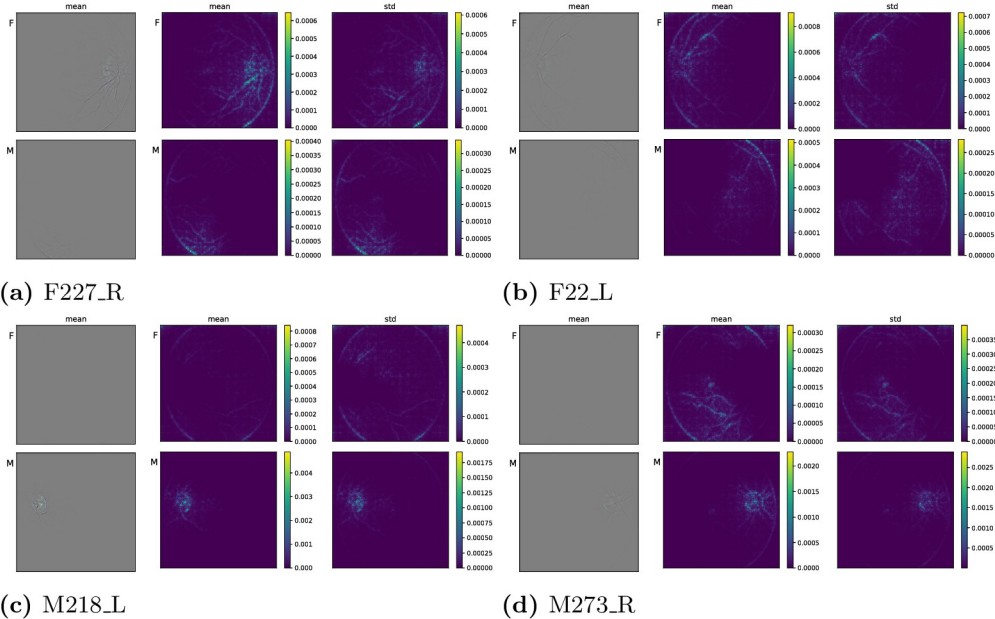

**Fig 6. Mean Guided Grad-CAM activations for Female (top row) and Male (bottom row) fundus images.** (a) F227_R, (b) F22_L, (c) M218_L, (d) M273_R.

In Fig 6a, the top row shows the pixel-wise summary statistics for GGCAM visualizations that were created to interpret image regions supporting a "Female" class label; the bottom row, "Male". Generally, in the top row, the image regions with greater values appear about the optic disk and the vasculature there, in particular. While there is some highlighting at the lower-right boundary of the eye, the standard deviation of the GGCAM-amps is also large there suggesting inconsistent activation there among the six models. In the bottom row, the image region with greatest average activation is in the lower-left of the image. There, the standard deviation is also large, again suggesting inconsistent activation across the six models. Notably, for this "Female" image, the mean activation is greater for the "Female" class label and tends to appear in more physiologically relevant regions of the fundus image (*i.e.*, about the optic disk and nearby vasculature; *cf.* [32, 35]).

Similar observations are present in Fig 6b. In particular, the average GGCAM activation for this "Female" fundus photo tends to be greater for the "Female" class label than the "Male" one. Furthermore, that activation is predominantly concentrated along the supratemporal vasculature and the optic disk. In contrast, the activation for the "Male" class label is predominantly concentrated along the border of the fundus image, which is not a physiological feature of the image.

For the "Male" fundus image corresponding to Fig 6c, the mean activation for the "Male" class label concentrates strongly on the optic disk and the vasculature there. Again, the mean activation is markedly stronger than for the opposite class label (*i.e.*, "Female"), and the activation for the opposite class label is predominantly concentrated on the border of the fundus photo.

## 4 Discussion

### 4.1 Training metrics

Despite the inherently stochastic nature of the training process, it can be seen (Fig 1) that each model run's performance is comparable to another's. That a model achieves similar

performance from realization to realization provides intuition that the training algorithm is behaving appropriately.

It is curious that the validation BCE score underwent an overall *increase* for all three data-bases. We observed this phenomenon consistently throughout our experiments. We believe it to be the result of complex interplay between the data distribution of retinal images, the complexity of the classification task, and the choice of loss function.

Our supplementary experiment (S1 Choice of loss in S1 File) compared two loss functions in an effort to shed light on this phenomenon: (1) binary cross entropy loss, and (2) an alternate loss function that is balanced in the way penalties change on the two halves of the curve straddling the threshold of 0.5. The results of this supplementary experiment indicate that both models show a tendency to generate a small number of incorrect but confident responses as training progresses. However, the phenomenon of increasing loss was no longer apparent when cross-entropy loss was replaced with the "balanced" loss function (see S1 Choice of loss in S1 File). A rigorous causal understanding of the behaviour that eventually yields a model producing outlier confident incorrect classifications for both categories is beyond the scope of the present work, and would make for an involved subject of further study.

## 4.2 Model-database duality

We observe that models perform better on the DOVS database in single-domain tasks. In particular, D1–5 achieve higher validation metrics during training (*cf*. Fig 1) and have greater test AUC (*cf*. Table 2). Furthermore, N1–6 and C1–6 attained greater AUC when evaluated on the DOVS-ii dataset, relative to D1–5 on either ODIR-N or ODIR-C (*cf*. Table 2). In fact, such results may initially seem surprising: shouldn't the higher-scoring models be more likely to perform well when adapted to new datasets? In fact, we believe this behaviour underscores how underlying structure in the data is critical to model performance. The ODIR database [40] was obtained from many different hospitals and medical centres using various cameras (produced by different companies), resolutions and image qualities. The effects of this variation are two-fold. First, this variation increases the complexity of the ODIR database, likely increasing the difficulty of the learning task. This foreshadows lower scores like those seen in Single-domain results. On the other hand, when using a sufficiently complex model, training on a dataset of greater complexity has the opportunity to produce a more robust model. This point of view is consistent with N1–6 and C1–6 achieving higher scores on the DOVS-ii database than D1–5 did on ODIR-N or ODIR-C. This is also consistent with the observation that D1–5 typically achieve higher scores when evaluated on the ODIR-C dataset than on the ODIR-N dataset.

In Domain adaptation results, we observed that models N1–6 and C1–6 attained test AUC scores above 0.6 (see Table 4c and 4d, respectively), while none of the models D1–5 attained this (arbitrarily chosen) threshold (Table 4a), nor did E1–10, E* (Table 4b). We offer the following *ex post facto* justification for this observation, which supports the description provided in the previous paragraph. Namely, the models fine-tuned on a relatively more complex data-set were able to generalize better to unfamiliar data than were the models developed using a more uniform database (DOVS).

Importantly, in Fig 5, we observed two interesting facts about ODIR-N *vs*. ODIR-C that are suggestive about the role of image quality in dataset curation and model development. First, compared to ODIR-N, models trained on ODIR-C generally achieved higher adapted AUC scores, even though their test AUC scores were uniformly lower. Second, D1–5, E1–10 and E* achieved better adapted AUC scores when tested on ODIR-C than when tested on ODIR-N. Together, these two observations suggest the importance of clean data when curating datasets

for model development; and for ensuring efficacy in domain adaptation tasks (such as deploying a developed model in a new clinical setting).

### 4.3 Ensembling

In this work, we investigated an ensemble generated by averaging model beliefs. We found that model averaging improved classifier performance, with only moderate increase in the computational complexity of the problem. Indeed, typically the development of a deep learning model involves several rounds of training to allow for hyperparameter tuning and changes to the model architecture. Thus, adequately performing models found during this development process are apt candidates to comprise an ensemble classifier.

Several interesting avenues present themselves for further study. For instance, we do not investigate how the ensemble could be tuned to produce improved results. For example, the ensemble could have been computed as a weighted sum of the individual networks, whose weights were determined by the validation AUC of each model; or through a voting procedure. Alternatively, the tuning procedure could have been learned during a second round of training. Such approaches are typically more suitable when larger databases are available. Finally, we do not examine dependence of AUC on $\ell$ or $L$. This subject would be an involved investigation that would be interesting to explore in future work.

### 4.4 Visual explanations for (mis)classification

We observed in Visual explanations for DOVS-ii classifiers that mean GGCAM behaviour for a matching class label (*e.g.*,"Female" image, "Female" class label) tends to have activation in image regions that have physiologically salient components (*e.g.*, vasculature and/or optic disk), some of which have previously been identified as potentially relevant markers for sex determination from fundus images [32]. Moreover, in contrast, mean GGCAM activation for an opposite class label (*e.g.*, "Female" image, "Male" class label) tends not to have activation in physiologically salient image regions (*e.g.*, at the border of the background image and the fundus). Generally, we also observe (albeit from a small number of examples) that the activation pattern for a "Female" image and "Female" class label is more diffuse and focused on vasculature, in contrast to "Male" images and "Male" class label, in which the activation pattern tends to be concentrated predominantly on and about the optic disk.

However, we also observe from individual GGCAM visualizations (see S1 GGCAM plots in S1 File) that the perceived quality of activation appears not always to correlate with the class label predicted by the model. Indeed, it is frequently the case that a model that misclassified a fundus image still produces a GGCAM for the *correct* class label that has physiologically relevant activation patterns (bearing similarity to ones of other models that gave a correct classification). Of the four fundus images considered, this phenomenon is perhaps most pronounced for M218_L. Indeed, half of the models misclassified this image, but all of them produced similar GGCAMs for both class labels. Thus, we believe it could be an interesting subject of future study to determine how visual explanations can be used for robustifying models against misclassification.

## 5 Conclusion

In this work we have exhibited how deep learning models can achieve super-human performance on challenging retinal image analysis tasks even when only small databases are available. Two previous studies have reported successful classification of sex from retinal fundus images using deep learning, though this was achieved via use of much larger datasets in each case: over 80000 images in Korot et al. [55] and over 1.5 million images in Poplin et al. [7].

Further, we have demonstrated that deep learning models for retinal image analysis can succeed at transductive transfer learning when applied to a similar task on a different data distribution. Finally, we have exhibited how the model development process lends itself to the creation of ensemble classifiers composed of deep learning models, whose performance can exceed that of its constituent models.

Data presents as the primary bottleneck in machine learning applications of medical image analysis, with reductions in dataset size often associated with reductions in model performance. In this proof-of-concept study, we have shown that small datasets can be leveraged to obtain meaningful performance in the context of retinal fundus image classification. Our study has revealed several important considerations when data set size is a limitation, such as use of transfer learning in model development, and use of ensembling to maximize classification performance. These results have also highlighted maintaining strict image quality criteria as a strategy to improve generalization in applications that involve domain adaptation. These findings move us one step further in the democratization and applicability of deep learning in medical imaging.

## Supporting information

**S1 File.**
(ZIP)

**S1 Fig.**
(PDF)

**S2 Fig.**
(PDF)

**S3 Fig.**
(PDF)

**S4 Fig.**
(PDF)

**S5 Fig.**
(PDF)

**S6 Fig.**
(PDF)

**S7 Fig.**
(ZIP)

**S8 Fig.**
(ZIP)

## Author Contributions

**Conceptualization:** Aaron Berk, Özgür Yılmaz, Ipek Oruc.

**Data curation:** Aaron Berk, Gulcenur Ozturan, David Maberley, Ipek Oruc.

**Formal analysis:** Aaron Berk, Özgür Yılmaz.

**Investigation:** Aaron Berk, Özgür Yılmaz, Ipek Oruc.

**Methodology:** Aaron Berk, Özgür Yılmaz, Ipek Oruc.

**Project administration:** David Maberley, Özgür Yılmaz, Ipek Oruc.

**Resources:** Aaron Berk.

**Software:** Aaron Berk.

**Supervision:** Özgür Yılmaz, Ipek Oruc.

**Validation:** Aaron Berk, Parsa Delavari.

**Visualization:** Aaron Berk, Parsa Delavari.

**Writing – original draft:** Aaron Berk, Parsa Delavari, Ipek Oruc.

**Writing – review & editing:** Aaron Berk, Özgür Yılmaz, Ipek Oruc.

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
