## [Decision Letter · Decision Letter 0]

1 Dec 2022

PONE-D-22-19392Learning from few examples: Classifying sex from retinal images via deep learningPLOS ONE

Dear Dr. Berk,

Thank you for submitting your manuscript to PLOS ONE. After careful consideration, we feel that it has merit but does not fully meet PLOS ONE’s publication criteria as it currently stands. Therefore, we invite you to submit a revised version of the manuscript that addresses the points raised during the review process.

We look forward to receiving your revised manuscript.

Kind regards,

Nguyen Quoc Khanh Le

Academic Editor

PLOS ONE

Journal Requirements:

^2. Please note that PLOS ONE has specific guidelines on code sharing for submissions in which author-generated code underpins the findings in the manuscript. In these cases, all author-generated code must be made available without restrictions upon publication of the work. Please review our guidelines at https://journals.plos.org/plosone/s/materials-and-software-sharing#loc-sharing-code and ensure that your code is shared in a way that follows best practice and facilitates reproducibility and reuse.^

Reviewers' comments:

Reviewer's Responses to Questions

**Comments to the Author**

1. Is the manuscript technically sound, and do the data support the conclusions?

Reviewer #1: Yes

Reviewer #2: Yes

2. Has the statistical analysis been performed appropriately and rigorously? 

Reviewer #1: Yes

Reviewer #2: Yes

3. Have the authors made all data underlying the findings in their manuscript fully available?

Reviewer #1: Yes

Reviewer #2: Yes

4. Is the manuscript presented in an intelligible fashion and written in standard English?

Reviewer #1: Yes

Reviewer #2: Yes

5. Review Comments to the Author

Reviewer #1: 1. Title – few-shot learning is another research field of AI. The authors used more than 1000 fundus images to train the networks. In the literature, this volume has not been considered as “few examples”.

2. The authors should read “Sex judgment using color fundus parameters in elementary school students, Graefe's Archive for Clinical and Experimental Ophthalmology volume 258, pages2781–2789 (2020)”. They found several patterns of fundus photography for determining sex. Very important paper in this field.

3. The results of deep learning classifiers should be explained using Grad-CAM or occlusion maps. The authors should report the patterns of males and females. Additionally, comparison with other studies (Sex judgment using~ and Predicting sex from retinal fundus~).

4. Fig 1 – the concept of back-propagation is not necessary for the academic readers. Too basic.

5. According to the previous studies, it is well-known that deep learning can determine sex via fundus photography. Please highlight the novelty and academic contribution of this study.

6. Please clarify the demographic information of the participants. Age and ethnicity.

7. Transfer learning or fine-tuning pretrained networks are not novel. The authors show what they did for few-shot or small-dataset-based learning. There are some few-shot learning articles: Feasibility study to improve deep learning in OCT diagnosis of rare retinal diseases with few-shot classification, Med Biol Eng Comput. 2021 Feb;59(2):401-415; Low-Shot Deep Learning of Diabetic Retinopathy With Potential Applications to Address Artificial Intelligence Bias in Retinal Diagnostics and Rare Ophthalmic Diseases, JAMA ophthalmology, 2020;

Reviewer #2: The explanation are clear and understandable. All References are relevant. Representation of Pictorial statistical information can be made easier to understand. Overall the proposed work is good and accepted for publication.

6. PLOS authors have the option to publish the peer review history of their article (what does this mean?). If published, this will include your full peer review and any attached files.

Reviewer #1: No

Reviewer #2: **Yes: **RAMASUBRAMANIAN BHOOPALAN

---

## [Author Response · Author response to Decision Letter 0]

21 Feb 2023

Please refer to our Response to the referees' comments.

---

## [Decision Letter · Decision Letter 1]

11 May 2023

PONE-D-22-19392R1Learning from small data: Classifying sex from retinal images via deep learningPLOS ONE

Dear Dr. Berk,

Thank you for submitting your manuscript to PLOS ONE. After careful consideration, we feel that it has merit but does not fully meet PLOS ONE’s publication criteria as it currently stands. Therefore, we invite you to submit a revised version of the manuscript that addresses the points raised during the review process.

We look forward to receiving your revised manuscript.

Kind regards,

Nguyen Quoc Khanh Le

Academic Editor

PLOS ONE

Journal Requirements:

Reviewers' comments:

Reviewer's Responses to Questions

**Comments to the Author**

1. If the authors have adequately addressed your comments raised in a previous round of review and you feel that this manuscript is now acceptable for publication, you may indicate that here to bypass the “Comments to the Author” section, enter your conflict of interest statement in the “Confidential to Editor” section, and submit your "Accept" recommendation.

Reviewer #1: All comments have been addressed

2. Is the manuscript technically sound, and do the data support the conclusions?

Reviewer #1: Yes

3. Has the statistical analysis been performed appropriately and rigorously? 

Reviewer #1: Yes

4. Have the authors made all data underlying the findings in their manuscript fully available?

Reviewer #1: Yes

5. Is the manuscript presented in an intelligible fashion and written in standard English?

Reviewer #1: Yes

6. Review Comments to the Author

Reviewer #1: The authors well addressed my concerns. The manuscript has been improved after revision.

Fig 6 needs the original fundus images.

7. PLOS authors have the option to publish the peer review history of their article (what does this mean?). If published, this will include your full peer review and any attached files.

Reviewer #1: No

---

## [Author Response · Author response to Decision Letter 1]

27 Jun 2023

Please see our Response to the referees' reports.

---

## [Editor Report · Decision Letter 2]

14 Jul 2023

Learning from small data: Classifying sex from retinal images via deep learning

PONE-D-22-19392R2

Dear Dr. Berk,

We’re pleased to inform you that your manuscript has been judged scientifically suitable for publication and will be formally accepted for publication once it meets all outstanding technical requirements.

Kind regards,

Nguyen Quoc Khanh Le

Academic Editor

PLOS ONE
---

## [Editor Report · Acceptance letter]

24 Jul 2023

PONE-D-22-19392R2 

Learning from small data: Classifying sex from retinal images
via deep learning 

Dear Dr. Berk:

I'm pleased to inform you that your manuscript has been deemed suitable for publication in PLOS ONE. Congratulations! Your manuscript is now with our production department. 

Kind regards, 

on behalf of

Dr. Nguyen Quoc Khanh Le 

Academic Editor

PLOS ONE